# Bayesian generative models can flag performance loss and bias

## Abstract

Generative models are popular for medical imaging tasks such as anomaly detection, feature extraction, data visualization, or image generation. Since they are parameterized by deep learning models, they are often sensitive to distribution shifts and unreliable when applied to out-of-distribution data, creating a risk of, e.g. underrepresentation bias. This behavior can be flagged using uncertainty quantification methods for generative models, but their availability remains limited. We propose SLUG: A new UQ method for VAEs that combines recent advances in Laplace approximations with stochastic trace estimators to scale gracefully with image dimensionality. We show that our UQ score – unlike the VAE's encoder variances – correlates strongly with reconstruction error and racial underrepresentation bias for dermatological images.

## 1 Introduction

Variational Autoencoders (VAEs) stand out as one of the most widely used generative models in medical imaging because of their capacity to learn semantic, low-dimensional latent spaces. VAEs are often used to analyze and manipulate key characteristics of high-dimensional data, e.g. for data visualization [3], data generation [8], and anomaly detection [14].

Despite these advantages, generative models are parameterized with modern Deep Neural Networks (DNNs), which struggle with out-of-distribution (OOD) [13]. To tackle OOD performance in predictive models, uncertainty quantification (UQ) has emerged as an important tool [9], where the prediction is endowed with an associated uncertainty. This has proven useful for detecting silent failures and ensuring that unexpected outcomes do not occur. While UQ techniques have been proposed and studied with promising success for discriminative models [1], their applicability to generative models such as VAEs is underexplored. The adoption of a Bayesian approach can help address this issue; however, current Bayesian generative models tend to be computationally expensive, difficult to tune, or rely on uncorrelated posterior approximations [12, 2].

**We propose** a novel epistemic UQ method for VAEs building on the *Sketched Lanczos Uncertainty (SLU)* algorithm recently proposed for discriminative models [11]. SLU computes a rank-$k$ approximation of the *generalized Gauss-Newton (GNN)* matrix, which captures the epistemic uncertainty according to a Laplace approximation [7]. However, SLU scales quadratically with the output dimension, which is intractable in image-generative models. Our proposed *Sketched Lanczos Uncertainty Global (SLUG)* measure overcomes this challenge using scalable stochastic trace estimators [6] to produce a per-image score.

**We demonstrate SLUG's ability** to detect underrepresentation bias in dermatological images. It is well known that dark skin tones are severely underrepresented in public datasets and that DNNs-based

Submitted to 39th Conference on Neural Information Processing Systems (NeurIPS 2025) Workshop: Medical Imaging meets EurIPS: MedEurIPS 2025. Do not distribute.

35  systems tend to reproduce and amplify this bias [5, 10]. Our experiments show that SLUG strongly
36  correlates with the performance of the VAE and can serve to flag both bias and performance loss.

37  In short, **we contribute** a novel UQ method for VAEs that capture out-of-distribution data and
38  demonstrate its utility in flagging errors and underrepresentation bias using two publicly available
39  real-world dermatology datasets: Fitzpatrick17k [5], and PASSION [4].

## 2  Method

41  Given the VAE $f_{\phi,\theta}$ with encoder and decoder parameters $(\phi, \theta) \in \mathbb{R}^p$, we define its Jacobian
42  $\mathbf{J}_{\{\phi,\theta\}}$ with respect to parameters and construct the Generalized Gauss-Newton (GGN) matrix as
43  $\mathbf{G}_{\{\phi,\theta\}} = \sum_{i=1}^{n} \mathbf{J}_{\{\phi,\theta\}}(x_i)^T \mathbf{H}(x_i) \mathbf{J}_{\{\phi,\theta\}}(x_i)$, where $\mathbf{H}(x_i)$ is the Hessian of the loss w.r.t. to the
44  neural network output and $\{x_i \in \mathbb{R}^{W \times H \times C}\}_{i=1}^{n}$ the training set.

45  The GGN commonly appears as the inverse covariance of the *linearized Laplace approximation (LLA)*
46  to the true posterior [7]. Currently, LLA is the most promising Bayesian posterior approximation [7],
47  but it is, unfortunately, intractable for generative models as its computational cost scales quadratically
48  with the generated data dimension, specifically $\mathcal{O}((WHC)^2 p)$ for a network with $p$ parameters.

49  Recently, Miani et al. [11] developed a sketching-based algorithm to evaluate the associated predictive
50  uncertainty, which scales logarithmically with $p$. The resulting *Sketched Lanczos Uncertainty (SLU)*
51  algorithm, however, still scales quadratically with the image dimension, making it impractical for
52  VAEs. Our approach extends SLU to scale gracefully to large images.

### 2.1  Scaling to VAEs: Sketched Lanczos Uncertainty Global score (SLUG)

54  We based our proposed score on the SLU algorithm [11]. Let $\mathbf{U}$ denote the matrix containing the
55  leading eigenvectors of the GGN, then the SLU approximates the predictive variance of the linearized
56  Laplace approximation with $\mathbf{I} - \mathbf{U}\mathbf{U}^\top$ is covariance, i.e., for a specific pixel $(w, h, c)$,

$$\text{SLU}_{w,h,c}(x) = e_{whc} \mathbf{J}_{\theta^*}(x)(\mathbf{I} - \mathbf{U}\mathbf{U}^\top)\mathbf{J}_{\theta^*}(x)^\top e_{whc}^\top, \tag{1}$$

57  where $e_{whc}$ is a one-hot encoding vector that selects the pixel position to compute the uncertainty.

58  SLU approximates this predictive uncertainty using several tricks from randomized numerical linear
59  algebra. Unfortunately, even SLU does not scale to neural networks with high-dimensional outputs
60  like those in generative models. Producing one predictive variance per generated pixel requires
61  $\mathcal{O}(WHC)$ SLU invocations, which is practically prohibitive. Our main interest is in measuring
62  a scalar uncertainty score for a generated image, and we choose the sum of per-pixel predictive
63  variances, which we denote the *Sketched Lanczos Uncertainty Global (SLUG)* score. We use a
64  stochastic trace estimator [6] to estimate the SLUG score,

$$\text{SLUG}(x) = \sum_{w,h,c} \text{SLU}_{w,h,c}(x) \tag{2}$$

$$= \text{Tr}(\mathbf{J}_{\theta^*}(x) (\mathbf{I} - \mathbf{U}\mathbf{U}^\top) \mathbf{J}_{\theta^*}(x)^\top) \tag{3}$$

$$\approx \frac{1}{S} \sum_{s=1}^{S} \epsilon_s \mathbf{J}_{\theta^*}(x) (\mathbf{I} - \mathbf{U}\mathbf{U}^\top) \mathbf{J}_{\theta^*}(x)^\top \epsilon_s^\top, \tag{4}$$

65  where $\epsilon_s \sim \mathcal{N}(0, \mathbf{I})$. This can be implemented using only $S$ invocations of SLU.

## 3  Experiments and results

67  **Datasets.**  In our experiments, we use two dermatology datasets. The VAE is trained on the
68  **Fitzpatrick17k** dataset [5], which includes 16,577 images labeled with Fitzpatrick skin types (FST).
69  To evaluate the impact of skin tone representation in the training data, we create three subsets of
70  1,668 images each: **Dataset A – Light** (100% FST 1–2), **Dataset B – Mixed** (50% FST 1–2 and 50%
71  FST 5–6), and **Dataset C – Black** (100% FST 5–6). Two separate test sets of 512 images each are
72  sampled for lighter and darker skin tones. For external validation and to assess model bias, we use
73  the **PASSION** dataset [4], which contains 4,901 dermatology images from Sub-Saharan countries
74  with darker skin tones (FST 3–6).

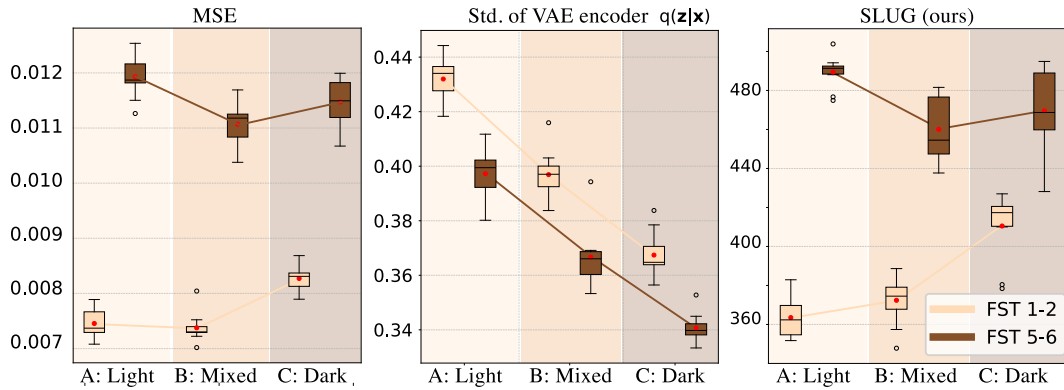

Figure 1: On Fitzpatrick17k, the performance on light and dark skin tones changes with their representation. The VAE encoder uncertainty is a poor indicator, while SLUG follows performance across groups and training scenarios.

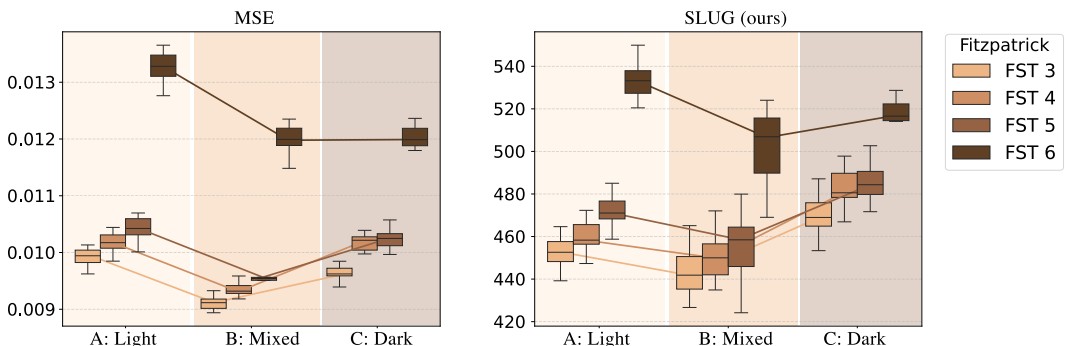

Figure 2: On the external PASSION dataset, we see again how reduced MSE is flagged by increased SLUG uncertainty across dark skin tone groups.

**Racial bias.** Fig. 1 shows that SLUG effectively captures racial bias in the Fitzpatrick17k dataset—unlike the VAE's latent uncertainty. We also evaluate the trained models in Fitzpatrick17k on the external PASSION dataset. The results reveal a consistent pattern: performance degrades as skin tone darkens (see Fig. 2). Notably, our SLUG score also captures this racial bias in the external dataset, highlighting its utility as a metric to flag bias.

## 4  Conclusion

This work highlights the urgent need for precise and scalable UQ for generative models. Despite the widespread use of generative AI, we still lack a reliable mechanism to ensure its trustworthiness. **We demonstrate that epistemic UQ can warn of performance loss, and detect bias**.

We expect that these results will motivate the recognition of epistemic uncertainty as an essential tool for generative models. While our SLUG method for VAEs can capture bias and performance loss in dermatological images, it will be valuable to see how epistemic uncertainty and bias interact in other large generative models, such as diffusion models.

## 5  Potential Negative Impact Statement

This paper studies the relationship between epistemic uncertainty and racial bias in dermatology using generative models, evaluated on the Fitzpatrick17k and a Sub-Saharan dataset. The analysis is limited to racial categories present in these datasets and does not consider other sources of bias, such as gender, age, or data origin. These unexamined factors may also influence uncertainty and model performance.

The models are evaluated on curated datasets and not tested under out-of-distribution or real-world clinical conditions. As a result, their robustness and reliability in practice remain unclear. Further evaluation in real-world and OOD settings is necessary to assess clinical safety.

While the method proposed may help identify racial bias, they do not prevent misuse or biased deployment. Ensuring fairness in clinical applications requires broader efforts, including careful validation, and responsible use by practitioners and developers.

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
