# OpenReview forum: "Bayesian generative models can flag performance loss and bias"
_EurIPS.cc/2025/Workshop/MedEurIPS — EurIPS 2025 Workshop MedEurIPS Submission_

### Official Review · Reviewer_jdd7 · 2025-10-24
**Well-executed work with clear contributions**

**Rating:** 8
**Confidence:** 4

**Review:**

The paper presents an epistemic uncertainty quantification method for VAEs. The proposed score combines the Sketched Lanczos Uncertainty algorithm with a stochastic trace estimator to make the computational cost more tractable with respect to the output image dimensions. Results on two public dermatology datasets show that the proposed score can reliably uncover high VAE reconstruction error and even under-representation bias in the data. Overall, the paper is clear, the method is somewhat novel, and the work should be of interest to the workshop’s audience. To strengthen their work, I recommend that the authors discuss the implications of using Hutchinson’s trace estimator (i.e., high variance) and how this might affect their uncertainty estimates.

---

### Official Review · Reviewer_7v12 · 2025-10-31
**Review - Bayesian generative models can flag performance loss and bias**

**Rating:** 6
**Confidence:** 4

**Review:**

The authors propose an extension to SLU, which is an uncertainty estimation method, using stochastic trace estimations that makes it computationally tractable for larger models such as VAEs used in image generation methods. On Fitzpatrick17k they show that SLUG nicely correlates with the reconstruction performance.

Strengths:
- Flagging uncertainty in VAEs is an important idea, as it could reduce the impact of many biases affecting the use of these models. For example, many autoencoders trained on natural images are currently used for medical use cases in dataset generation.
- There is a good correlation between MSE reconstruction performance and SLUG.
- The authors compare their method to the standard deviation naturally computed by VAEs and show that it clearly outperforms it.

Weaknesses:
- In the introduction, the authors mention that the method could be applied to image generation use cases. This is a fantastic idea, if it works. However, the paper currently does not demonstrate this. It would be great to see a visual example illustrating where this method flags uncertainty. For example, in diffusion models, when a VAE trained on general images is used with medical images and fails to reconstruct them. I am not entirely convinced that this problem exists in the broader context of image generation.
- SLUG also seems to be highly class-dependent. Looking at Figure 1, it is clear that the impact of different skin types is much greater than the impact of the generated subset.

Overall the paper is interesting and shows promising results which is why I recommend acceptance.

---

### Decision · Program_Chairs · 2025-11-03

**Decision:**

Accept (Poster)

**Comment:**

Both reviewers agree that the paper is well-executed, methodologically sound, and relevant to the MedEurIPS community.